# A Beginner’s Guide to Cell Culture: Practical Advice for Preventing Needless Problems

**DOI:** 10.3390/cells12050682

**Published:** 2023-02-21

**Authors:** Sabine Weiskirchen, Sarah K. Schröder, Eva Miriam Buhl, Ralf Weiskirchen

**Affiliations:** 1Institute of Molecular Pathobiochemistry, Experimental Gene Therapy and Clinical Chemistry (IFMPEGKC), RWTH University Hospital Aachen, D-52074 Aachen, Germany; 2Electron Microscopy Facility, Institute of Pathology, RWTH University Hospital Aachen, D-52074 Aachen, Germany

**Keywords:** contamination, mycoplasma, retrovirus, STR profiling, misidentification, cell authentication, conditional reprogramming

## Abstract

The cultivation of cells in a favorable artificial environment has become a versatile tool in cellular and molecular biology. Cultured primary cells and continuous cell lines are indispensable in investigations of basic, biomedical, and translation research. However, despite their important role, cell lines are frequently misidentified or contaminated by other cells, bacteria, fungi, yeast, viruses, or chemicals. In addition, handling and manipulating of cells is associated with specific biological and chemical hazards requiring special safeguards such as biosafety cabinets, enclosed containers, and other specialized protective equipment to minimize the risk of exposure to hazardous materials and to guarantee aseptic work conditions. This review provides a brief introduction about the most common problems encountered in cell culture laboratories and some guidelines on preventing or tackling respective problems.

## 1. Introduction

Cell culture experiments are widely used in biomedical research, regenerative medicine, and biotechnological production. Due to restrictions on the use of laboratory animals by animal protection laws and the strict implementation of the 3Rs (**R**eplacement, **R**eduction, and **R**efinement) formulated by William Russell and Rex Burch to improve the welfare of animals, it can be expected that the general use of cell lines will further increase during the next years to substitute animal-based research [1]. However, it should be noted that cell culture experiments, when not properly conducted, are prone to errors. Therefore, it is essential that cell culture studies are performed with good cell culture practice (GCCP) to assure the reproducibility of in vitro experimentation [2].

In particular, inter- and intra-specific cross-contamination and cell misidentification, genetic drift, contamination with bacteria, fungi, yeast, viruses, or chemicals, and lack of quality control testing are widespread fatal cell culture problems that contaminate the literature with false and irreproducible results [3]. Rough estimates suggest that the number of published papers that used problematic cell lines is about 16.1% [4]. Moreover, the International Cell Line Authentication Committee (ICLAC) lists 576 misidentified or cross-contaminated cell lines in its latest register released in June 2021 [5].

Although it is hard to estimate how much misguided articles are actually affected, there is still an urgent need to better sensitize scientists to this problem. Furthermore, biosafety and ethical aspects are not in the public awareness or are even ignored when working with cell lines. Exemplarily, several continuous growing cell lines were established by transformation with the Simian virus 40 (SV40) large T-antigen (SV40T) or other agents with oncogenic potential, immortalized by introducing telomerase reverse transcriptase (TERT) activity, derived from genetically modified animals, or by novel technologies such as CRISPR/Cas9 gene editing [6,7,8]. Consequently, many cell lines need to be classified as genetically modified cell lines (GMCLs) that need sufficient safety attention.

During the last decades, scientists have established many guidelines on good cell and tissue culture practice (GCCP) that provide continuously updated guidance on the main principles to consider when performing cell culture. The GCCP guidelines highlight issues of quality management, background on culture systems, documentation and reporting, general safety instructions, information about education and training, and ethical issues associated with the performance of cell culture experiments [2,9]. Moreover, these expert documents try to promote the harmonization, rationalization, and standardization of laboratory practices including manufacture and testing to foster the compliance of researchers’ work with laws, regulations, and ethical principles. In addition, these guidelines provide extensive information about essential, beneficial, and useful additional equipment for setting up and furnishing a cell culture laboratory environment with a focus on national and international agreed standards. These guidelines are rather complex and comprehensive, so they will be of interest primarily to trained researchers, who have extensive experience in performing cell culture experiments for many years.

In this article, some general aspects of working with cell lines are discussed on a more simplified level. In particular, tests for the authentication of cell lines, potential cell culture contaminants, and brief information about ethical issues and biological safety guidelines for use of cell lines in biomedical research are summarized. As such, the information provided should be useful for those that will start to conduct cell culture experiments.

## 2. Classification of Cell Culture Types

Cell lines can be roughly classified into three groups, namely (i) finite cell lines, (ii) continuous cell lines, also known as immortalized or indefinite cell lines, and (iii) stem cell lines [2]. Finite cell lines are normally derived from primary cultures and have slow growth rates. As such, they can be grown for a limited number of cell generations in culture before finally undergoing aging and senescence, a process that is indicated by loss of the typical cell shape and enrichment of cytoplasmic lipids. Importantly, finite cell lines are contact-inhibited and arrested in the G_0_, G_1_, or G_2_ phase after forming monolayers [10].

In contrast, continuous growing cell lines are typically obtained from transformed or cancerous cells and divide rapidly and achieve much higher cell densities in culture than finite cell lines. In some cases, these cell lines exhibit aneuploidy (i.e., one or more chromosomes being present in greater or lesser number than the others) or heteroploidy (i.e., having a chromosome number that is other than a simple multiple of the haploid number). They often can be grown under reduced serum concentrations, are not contact-inhibited, and might form multilayers. Stem cells are an undifferentiated or partially differentiated pluripotent cell type originating from a multicellular organism. These cells can be extended to indefinitely more cells of the same type or alternatively can be triggered under the right conditions to produce cells with specialized functions. As such, they can act as a kind of multipotent precursor for many different cell types.

In all cases, the growth of cells from various sources requires an artificial but controlled environment, in which sometimes highly specialized media, supplements, and growth factors are needed for proper cell growth. A cell type can either grow adherent (attached to a surface) requiring a detaching agent for passaging, or alternatively can be free floating in suspension. Adherent cells can be further divided into fibroblast-like cells having an elongated shape and epithelial-like cells characterized by a polygonal shape. Similarly, each cell culture can have unique properties in regard to morphology, viability, doubling time, and genetic stability and their handling and maintenance may require different media, culture conditions, and additives or processing agents including antibiotics, detachment solutions, or surface coating for cell attachment [2]. Non-adherent or suspension cells grow either as single cells or as free-floating clumps in liquid medium that do not require enzymatic or mechanical dissociation during passaging. However, in some cases, these cells demand shaking or stirring for adequate gas exchange and proper growth. Typical examples of non-adherent cells are hematopoietic cell lines derived from blood, spleen, or bone marrow that proliferate without being attached to a substratum. Nevertheless, also some adherent cell lines can be adapted to grow in suspension, which allows for more manageable cell culturing at larger scales with higher yields in special applications [11,12]. In addition, compared to adherent cells, cells grown in suspension are generally easier to handle. Exemplarily, when adherent cells should be analyzed by analytical flow cytometry or fluorescence-activated cell sorting (FACS), the cells must first be detached from their substratum. However, enzymatic digestion or the usage of non-enzymatic cell dissociation buffers can result in the degradation of surface proteins, which might prevent their subsequent identification and cell separation in respective protocols. This makes it extremely challenging to use flow cytometry for phenotyping and characterization of adherent cells [13]. Trypsin, for example, is frequently used for detaching adherent cells. It time-dependently degrades most cell surface proteins by cleaving peptides after lysine or arginine residues that are not followed by proline [14]. This results in degradation of most surface proteins during cellular dissociation. Similarly, other enzymes such as extracellular matrix-specific collagenases, the serine protease elastase cleaving the peptide bond of C-terminal neutral, non-aromatic amino acid residues [15], the peptidase dispase hydrolyzing *N*-terminal peptide bonds of non-polar amino acid residues [16], and many other detachment agents provoke the significant break down of proteins.

Therefore, several milder enzyme mixtures such as Accutase and Accumax or non-enzymatic cell dissociation reagents such as a mixture of ethylenediaminetetraacetic acid (EDTA) and nitrilotriacetic acid (NTA) chelating divalent metal cations have been introduced for routine cell passaging and manipulation of sensitive cells. These formulations are less toxic and preserve most epitopes for subsequent flow cytometry analysis [17,18].

More recently, the culturing of cells in a three-dimensional microenvironment has become the focus of researchers. These 3D cell cultures are either produced by culturing cells within a defined scaffold such as hydrogel or polymeric materials derived from extracellular matrix proteins or agarose or as self-assembly systems in which the cells grow in clusters or spheroids. It is well-accepted that these in vitro cell models offer the possibility to study cellular reactions in a closed system that better resembles the physiological situation than cell culture technologies that rely on two dimensions. As such, these models are particularly interesting for those studying aspects of cell-to-cell interactions, tumor formation, drug discovery, stem cell research, and metabolic interactions [19]. In comparison to 2D systems, 3D models have the potential to completely change the way drug efficacy testing, disease modeling, stem cell research, and tissue engineering research take place [19]. Finally, these systems will substantially decrease the use of laboratory animals in some research areas, which is a key aspect of the 3R principle [1].

## 3. Culture Media

Proper cell culture media are critical in the maintenance and growth of cell cultures and to allow the reproducibility of experimental results. Some cells additionally need non-essential amino acids (alanine, asparagine, aspartic acid, glutamic acid, glycine, proline, and serine) for effective growth and the reduction of the metabolic burden of cells.

The most common standard media used to preserve and maintain the growth of a broad spectrum of mammalian cell types are, for example, Dulbecco’s modified Eagle medium (DMEM) and Roswell Park Memorial Institute (RPMI) media. Typically, these media contain carbohydrates, amino acids, vitamins, salts, and a pH buffer system (Table 1).

Common media such as DMEM are available in a ready-to-use liquid form or alternatively in powdered media for easier storage and a longer shelf life. Moreover, many media can be obtained with different glucose concentrations (low or high glucose) as well as in formulations with and without L-glutamine or alternatively with stabilized glutamine. Finally, they are sold with or without a pH indicator such as phenol red.

Importantly, basal media typically contain no proteins, lipids, hormones, or growth factors. Therefore, these media require supplementation with fetal bovine serum (FBS), or often referred to as fetal calf serum (FCS), commonly at a concentration of 5–20% (*v*/*v*). FBS is obtained from the blood of fetuses of healthy, pre-partum bovine dams. The final serum is depleted of cells, fibrin, and clotting factors by centrifugation of the clotted blood. It should be noted that FBS from different sources might differ in growth factor and hormone profiles, virus content, endotoxin load, osmolality, total protein and metal content, sugars, and final processing (e.g., filtration, testing for potential contaminations). Therefore, most scientists prefer to buy traceable FBS batches with reliable lot-to-lot consistency to obtain reproducible results during experimentation.

In addition, newborn calf serum (NBCS) obtained from calves less than 20 days of age, calf bovine serum (CBS) sourced from calves aged between 3 weeks to 12 months, and adult bovine serum (ABS) isolated from adult cows more than 12 months old are frequently used to supplement cell culture media. Some researchers routinely heat-inactivate the serum at 56 °C for 15–30 min to inactivate the complement and to destroy potential bacterial contaminants. However, in most cases this is not necessary and should be omitted because heat inactivation also reduces the concentration or biological activity of growth factors that are required for proper cell growth.

However, it should be noted that serum-containing media has a number of disadvantages. Serum is complex, has an indefinite composition leading to batch-to-batch variation, increasing the risk of contamination, and the use of serum is commonly associated with ethical concerns in terms of avoiding the suffering of fetuses and animals [21,22]. Therefore, the development of serum-free media (SFM) has become a research hotspot during the last decades [21]. In principle, SFM can be divided into five types, namely (i) common SFM, (ii) xeno-free medium containing human-source but no animal components, (iii) animal-free medium, (iv) protein-free medium, and (v) chemically defined medium [21,23]. All these media contain key components (e.g., energy sources, vitamins, amino acids, lipids, trace elements, and inorganic salt ions) and are often enriched with special supplements such as anti-shear protectants, nucleic acids, and other ingredients that are required to improve the culture performance for certain cell types or applications [21]. Unfortunately, certain companies and suppliers of SFM often provide incomplete or no information at all about the composition of their media. Therefore, researchers already started a decade ago to install an online serum-free online database for the interactive exchange of information and experiences concerning SFM [22].

Biological contamination arising from bacteria, yeast, fungi, and mycoplasma can be better prevented by the addition of antibiotics and anti-mycotics to cell culture media. Most of them act by either inhibiting cell-wall synthesis (e.g., penicillin), interfering with membrane permeability (e.g., amphotericin B), or by inhibiting protein synthesis by preventing the assembly of the bacterial initiation complex between mRNA and the bacterial ribosome (e.g., streptomycin). However, the routine usage of antibiotics might develop slow growing persistent/resistant bacterial contaminants that may cause subtle alterations of cell differentiation and behavior [24]. In addition, antibiotics such as penicillin, streptomycin, and gentamycin can significantly alter gene expression and regulation and could modify the results of studies focused on drug response, cell regulation, and differentiation [25,26]. For example, a concise review has recently highlighted numerous publications that have shown the impact of antibiotics and antimycotics such as penicillin/streptomycin, gentamicin, and amphotericin B on in vitro properties of cells including proliferation, differentiation, survival, and genetic stability [27]. Similarly, a comprehensive literature search has found a number of reported side effects that are induced by different antibiotics, again supporting the notion that antibiotic-free culture media are recommended when possible to ensure the reliability and reproducibility of cell culture findings [28]. Consequently, researchers should avoid the permanent use of antibiotics in cell culture and should better try to implement strict aseptic working conditions to prevent bacterial contaminants in cell culture.

## 4. Phenol Red

Phenol red also known as phenolsulfonphthalein is the most frequent pH indicator in cell cultures. This water-soluble dye is a yellow zwitterion at low pH, while it changes to a red-colored anion or a fuchsia-colored di-anion at more basic conditions (Figure 1). Therefore, this dye has been used as an inert pH indicator dye in many tissue culture media to detect pH shifts, waste products of dying cells, or overgrowth of contaminants that typically cause an acidification of the medium. However, based on its structural resemblance to some non-steroidal estrogens, it has the capacity to bind to estrogen receptors with an affinity of 0.001% of that of estradiol, thereby stimulating the proliferation of estrogen receptor positive cells [20]. Thus, it is advisable to dispense phenol red during experimentation when working with estrogen-responsive cell systems.

## 5. Cell Contamination

Both biological and chemical agents might lead to contamination in cell cultures (Table 2). In most cases, slow cellular growth, change in morphology, fast change of pH in media, and elevated quantities of death or floating cells in the culture are the consequence. Therefore, cell cultures should be routinely screened for respective contaminants to prevent inconsistent results and other serious consequences. The most important contaminants are discussed in the following table.

### 5.1. Mycoplasma Contamination

Mycoplasmas are the smallest self-replicating organisms belonging to the bacterial class *Mollicutes*. They consist of a lipoprotein plasma membrane, ribosomes, and a genome consisting of a circular, double-stranded DNA molecule that ranges in size from 580 to 2200 kb [36] (Figure 2A). Mycoplasmas have limited biosynthetic capabilities and need to enter an appropriate host in which they multiply and survive for long periods of time [36]. Their tiny size (~0.1–0.2 µm) makes their identification impossible under a standard bright-field microscope, which is the reason why they are often go undetected in many laboratories. They lack a cell wall and are resistant to many common antibiotics that are used in cell cultures such as penicillin or streptomycin. Most important, mycoplasma contamination does not generate the turbidity that is characteristic for contamination by other bacteria or fungi.

Systemic studies have identified typical ways in which mycoplasmas can spread into cell culture. They include the introduction of mycoplasma cross-contamination from infected cultures, media, sera, or reagents that were obtained from other research laboratories or commercial suppliers. Moreover, the usage of non-sterile supplies, the infection by laboratory personnel who are carriers of mycoplasma, diffusion of mycoplasmas in incubators or hoods, contamination of cell cultures in liquid nitrogen tanks, transmission via airborne particles and aerosols, overuse of antibiotics, and improper sealing of culture dishes are other sources favoring mycoplasma contamination [37].

Although mycoplasma compete with the host cell for biosynthetic precursors and nutrients, the observed alterations in growth rates in affected cell cultures are often minimal, which is the reason why respective contamination is not readily detected. Nevertheless, mycoplasmas can extensively affect the host’s DNA, RNA, and protein metabolisms, impact intracellular amino acid and available ATP levels, modify cellular surface antigens, and can provoke fragmentation of DNA and other significant chromosomal alterations [37]. Consequently, regular testing and quick identification for such contaminants is highly crucial to prevent falsified research results, misleading publications, and the waste of research money. To enable this, a number of sensitive and specific tests were developed that allow the detection of respective infections, often without considering the origin or species. In the gold standard to detect mycoplasma contamination, a conditioned culture supernatant sample is added first to a liquid medium for mycoplasma culture and incubated after a few days on broth, agar, or indicator cells [37]. However, this method is time consuming and other faster detection methods were introduced. They include, among others, specific DNA staining by fluorochromes such as 4′,6-diamidino-2-phenylindole (DAPI) or Hoechst 33342 (Figure 2B), ELISA testing, RNA labeling with mycoplasma-specific ribosomal probes, enzymatic procedures, flow-cytometric methods, colorimetric or strip-based mycoplasma detection assays, sensitive PCR-based assays detecting the bacterial 16S rRNA, and sophisticated Fourier transform infrared (FITR) microspectroscopy methods [38,39,40,41,42]. The staining with fluorochromes is rather non-specific, while most of the other assays have high analytical sensitivity (i.e., the ability of the test to identify the contaminant) and specificity (i.e., the ability to measure mycoplasmas and not closely related non-mycoplasma species). In particular, commercially available kit systems that rely on conventional (or endpoint) PCR often have the capacity to simultaneously detect more than 70–90 mycoplasma species because the primer set included in these kits is most often designed to specifically target and amplify the highly conserved 16S rRNA coding region of the mycoplasma genome (Figure 2C).

Mycoplasmas can produce a virtually unlimited variety of effects in infected cultures [43]. Therefore, a wide spectrum of agents and methods for eradication of respective contaminants was developed. There are different physical, chemical, immunological, and chemotherapeutic treatments available to eliminate mycoplasma contaminants [43]. However, many of these methods are impractical because they are either time-consuming, require special equipment, capture only a limited number of mycoplasma species, or have an overall low efficiency. In the beginning, several antibiotics that suppress mycoplasma growth were introduced, but most of them were only moderately effective, had detrimental effects on eukaryotic cells, or had only a limited efficacy because of the development of resistance against respective agents. Nevertheless, several more selective and effective anti-mycoplasma agents are now available, which allow the elimination of mycoplasma infections in most cases already by one round of treatment (Table 3). In most cases, these removal agents either contain macrolides, tetracyclines, quinolones, or combinations thereof. The different quinolones included in the common compounds Baytril^®^, Ciprobay^®^, and Zagam^®^ have similar chemical structures (Figure 3). They exert their inhibitory activity by blocking bacterial nucleic acid synthesis through disrupting the bacterial topoisomerase II, inhibiting the catalytic activities of topoisomerase IV and DNA gyrase, and by causing breakage of bacterial chromosomes [44]. The macrolide Tiamulin included in BM-Cyclin acts by inhibiting protein synthesis by targeting the 50S subunit of the bacterial ribosome and is further a strong inhibitor of the peptidyl transferase [45]. Its activity against mycoplasma is significantly enhanced by tetracycline [46]. Other ready-to-use removal agents contain combinations of antibiotics and antimicrobial acting peptides or combinations of three different bactericidal components belonging to different antibiotic families.

The different bactericidal components have a high effectiveness in eliminating mycoplasma infections. Typically, these substances are already effective against all common mycoplasma strains when administered for two or three weeks [47]. Exemplarily, the complete eradication of mycoplasma in a rat cell line as assessed by electron microscopy is shown in Figure 4.

Nevertheless, treatment or prevention of mycoplasma infections with these products can provoke significant cytotoxic and genotoxic effects [48]. Minocycline, for example, was shown to induce Bcl-2, which accumulated in mitochondria and interacted with death-promoting molecules including Bax, Bak, and Bid, thereby protecting against cell death [49]. Similarly, Tiamulin inhibited growth and metastasis of human breast cancer cell line MDA-MB-231 and mouse breast cancer cell line 4T1 by blocking the activity of 5′-nucleotidase (CD73) that catalyzes the conversion of purine 5′ mononucleotides to nucleosides [50]. Therefore, the application of compounds used for the removal of *Mycoplasma* infections should be used carefully and well-considered because they have the potential to alter the outcome of experimental studies.

### 5.2. Contamination with Viruses

Compared with mycoplasma infections, viral contaminants in cell cultures present a more serious threat because of the difficulty in detecting them and the lack of methods of treating affected cell cultures [51]. Moreover, some viruses have the capacity to integrate their genome into the host cell, which in some cases results in the permanent production of new viral particles. This is a potential health risk for operators and might be the source for horizontal virus transmission into other cell lines. Accordingly, this has direct implications on the biological safety classification of an infected cell line. Unfortunately, generic tests for the systemic evaluation of viral contaminants are rather complex and undirected. Without precise knowledge of the virus authenticity, these detection methods are limited to electron microscopy and assays for retroviral reverse transcriptase [52,53]. In particular, the versatility of electron microscopy is an effective, universal, and unbiased means when an infectious viral agent is suspected [54]. This technique is suitable for obtaining high resolution images, thereby providing an immediate overview of the actual cell infection status and shape of the viruses. The observed morphology and size further allow the immediate preliminary classification of the viral type [55]. Exemplarily, mature retroviruses are generally spherical enveloped particles with an average diameter ranging between 100 and 200 nm, displaying a distinct morphology that differs between the six retrovirus genera [56]. Particularly, transmission electron microscopy (TEM) is suitable for direct visualization of viruses in biological samples without the need of prior assumptions about the infectious agent. Illustratively, the retroviral load of the continuous growing murine hepatic stellate cell line GRX that is widely used in hepatology research was first demonstrated by TEM [53,57]. After all these years, this cell line has still the capacity to produce large quantities of retroviral particles (Figure 5).

### 5.3. Chemical Contamination

Any non-living compound evoking undesirable effects in a cell culture needs critical attention. Even essential nutrients can have toxic effects when the concentration is high enough. Impurities can be introduced into cell cultures by media, sera, and water. However, plasticizers in plastic tubing, cell culture disks, and storage bottles, as well as free radicals formed in media by fluorescent or UV light or deposits on glassware and pipettes, can result in contaminations. In addition, impurities in the CO_2_ flow and residues from germicides or pesticides used to disinfect incubators or hoods can be critical. Most critical are endotoxins, which are derived from the outer cell membrane of most Gram-negative bacteria. These are composed of rather stable lipopolysaccharides (LPS) that are shed from bacteria and are released, in a much greater quantity, during the lysis of bacteria [58]. Consequently, the removal of pyrogens from glassware requires extensive heating at high temperatures, while these substances are rather resistant to autoclaving. Therefore, most commercially available ready-to-use cell culture media and supplements are prepared in such a manner that they contain no or rather low endotoxin levels. Furthermore, reliable producers of cell culture media and supplements provide certificates of analysis indicating the endotoxin levels in respective solutions or compounds.

Some culture conditions (e.g., low serum concentration, high light exposure) provoke the formation of free radicals that are highly reactive, potentially leading to DNA damage, protein cross-links, lipid hydroperoxides, and the induction of apoptosis [59]. Therefore, antioxidants such as ascorbic acid, *N*-acetyl-L-cysteine, or vitamins (vitamin E, vitamin A, vitamin C) with free radical-scavenger activities are often added to cell culture media to prevent oxidative stress and its consequences [60,61]. Alternatively, the trace element selenium that functions as a cofactor in antioxidant enzymes, such as glutathione peroxidase, can be added to the media to annihilate reactive peroxides [62].

Additionally, high concentrations of heavy metals such as lead, cadmium, and mercury are toxic to many cell types. Therefore, it is important that water and solvents used to prepare media or supplements are tested for heavy metals prior to use. Alternatively, pure water from a commercial vendor that provides a certificate of analysis should be used if the laboratory water is not suitable for cell cultures.

### 5.4. Inter- and Intra-Species Cross-Contamination

Contamination of cell lines with unrelated cells from the same species (intra-species contamination) or cells from another species (inter-species contamination) is a common and recurrent problem [33,34]. In particular, when the contaminant is a rapidly dividing cell line, it will overgrow and replace the original culture. If the contamination is undetected, this may result in unreliable and irreproducible findings that falsify the biomedical literature [3]. The problem of cell line cross-contamination has been known for decades, commencing with the controversies around HeLa cells in the 1960s [45,46]. A first synopsis published in 2010 that was drawn from 68 references listed 360 cross-contaminated cell lines [33]. Subsequently, in 2021, the International Cell Line Authentication Committee (ICLAC) that aims to bring researchers’ attention to this problem listed 576 misidentified cell lines from different species resulting from cross-contamination or other means [5] (Table 4). The sources for cross-contamination are manifold and include unwanted spreading via aerosols or unplugged pipettes, sharing media and reagents among different cell lines, and careless usage of conditioned medium [34].

Nowadays, different methods for determination of cell line cross-contamination are available. Inter-species cross-contamination can be detected by isoenzyme analysis, which utilizes electrophoretic binding patterns to examine slight differences from species to species in the structure and mobility of individual isoforms for a set of intracellular enzymes [63]. Intra-species cross-contamination of human cells can be identified by typing of the human lymphocyte antigen (HLA) locus, which is a complex of genes located on chromosome 6 encoding highly polymorphic cell-surface proteins involved in immune system regulation. Moreover, molecular serological methods, genetic tests using synthetic probes or primers, and sequence-based typing with direct DNA sequencing can discriminate between different HLA genotypes [64].

### 5.5. Cell Misidentification

Similar to cell cross-contamination, the misidentification of cell lines has resulted in thousands of misleading and erroneous papers [65]. Usage of illegible labels and mislabeling of cell culture vessels during routine manipulation evoked by lack of attention, high operator workload, or distractions during experimental work is the most straightforward cause of misidentification [66]. It is comprehensible that a contamination with a more rapidly dividing contaminant cell can rapidly overgrow the original culture in a few passages and provokes cell line misidentification at later steps. Such an overgrowth can result when re-feeding operations and manipulations of multiple cultures are conducted with the same pipette or at the same time, as a consequence of intentional co-cultivation of different cell types, or by a lack of awareness when working with feeder cells.

## 6. Short Tandem Repeat Profiling

Nowadays, the most common method to identify cross-contamination and cell misidentification is short tandem repeat (STR) profiling. This method can compare the number of allele repeats at specific loci in DNA between different samples. Although the respective allelic variants of these repeats are rather polymorphic, the number of alleles is very small. Therefore, multiple STR loci are analyzed simultaneously in a multiplex PCR assay for making different STR profiles effective for identification or discrimination purposes with a high discriminative statistical power.

In STR analysis, the amplified variable microsatellite regions obtained from the template DNA are separated on a genetic analyzer and subsequently analyzed with software that calculates the number of repeats at each variant site. Nowadays, effective and standardized STR panels are established for many species. In this regard, the Consortium for Mouse Cell Line Authentication that has established a multiplex PCR assay comprising 19 mouse STR markers is pioneering [66,67]. This multiplex PCR provides a unique STR profile for different mouse cell lines, including closely related cell lines. Representative chromatograms of four STR markers obtained for the widely used immortalized murine cell line AML12 are depicted in Figure 6.

Importantly, when comparing the 19 mouse STR markers between three different mouse cell lines, each cell line has a unique constellation that allows the unequivocal discrimination of each cell line from the others (Table 5).

Nowadays, databases are available in which primer information for the setup of STR testing for mouse, cat, dog, cattle, horse, men, and others are deposited [69]. Moreover, the Cellosaurus resource provides an incredible wealth of information and offers routines such as the STR Similarity Search Tool (Cell Line Authentication using STR, CLASTR) for comparing STR profiles [70].

## 7. Cell Line Alteration and Over-Passaging

Typically, immortalized cell lines are grown in the lab for many generations. However, a cell line cultured at high passage number or for prolonged times can show chromosomal duplications or rearrangements, mutations, and epigenetic changes [71]. This phenomenon is commonly known as genetic drift. Consequently, the morphology, proliferation rate, metabolic capacity, or general cell health can change dramatically, affecting experimental outcomes [72]. Therefore, the documentation of cell line passage number, which reflects the number of times the cells have been subcultured into a new vessel, is an important consideration when performing an experiment.

It was also reported that the passage number can increase the risk of viral contamination [73]. Furthermore, over-passaging of cells selects faster growing cells that in some cases show reduced secretion rates, carrier-mediated transport, and paracellular permeability, while having increased transcellular permeability [74]. Consequently, similar or even the same investigations performed in different laboratories might have completely different experimental outcomes when the passage number differs by hundreds of passages. Although there are no specific guidelines regarding the optimal passage range, common practice is to not use cells after 20 to 30 passages. Unfortunately, the precise knowledge of passage number is often not known, especially when the cells were obtained from a source other than a cell repository, which usually provides data on the cell passage number [75].

In some cases, researchers argue that even the passage number is imprecise because different laboratories may use different initial cell seeding densities or splitting rates during passing, which affect the number of times cells divide in cultures. Therefore, a formula for the calculation of precise population doubling level (PDL), which is synonymous with the cell generation time, was introduced, which is used particularly for primary cells. In the respective formula, the PDL, which is the total number of times the cells in the population have doubled since their primary isolation in vitro, is calculated as follows:

PDL = 3.32 (log X_e_ − log X_b_) + S, where X_b_ is the number of seeded cells at the beginning of the incubation time, X_e_ is the cell number at the end of the incubation time, and S is the starting PDL before splitting [75,76,77].

## 8. Biosafety Aspects in Working with Cells

Cell cultures may have the ability to cause harm to human health and the environment and need to be assigned to a biosafety level that takes into consideration a multitude of factors [78]. Before working with a cell line, it is necessary to have an accurate knowledge about these risks, taking into account the intrinsic properties, type of (genetic) manipulation, and the resulting biological hazard inherent with the respective cell line that may be significantly increased by contaminating pathogens. Although these estimates must be conducted on a case-by-case basis, there are some general guidelines that need to be followed. First, the closer the genetic relationship of a cell under investigation is to humans, the higher the risk is to humans because contaminating pathogens usually have a specific species barrier. Nevertheless, care should be taken because some contaminating organisms have the potential to cross the usual species barrier [79,80,81]. Second, the tumor-inducing potential of a cell line is strongly dependent on its origin. While, for example, epithelial and fibroblastic cells have a low tumor-inducing potential, hematopoietic cells have a significantly higher one [82]. Third, well-characterized cell lines that are already used in many laboratories for many years have an overall lower risk than uncharacterized continuous growing cell lines or primary cells. After identifying and evaluating potential risks, it is essential to define ways of avoiding or minimizing these risks by containment, restricting the movement of staff and equipment into and out of cell culture laboratories, working according to standard operating procedures (SOPs), avoiding formation of aerosols or splashes during working, regular cell culture training, and by the implementation and following of the general guidelines of good laboratory practice (GLP) [2,78]. In addition, vaccination against Hepatitis B virus is advisable when working with primary human cell cultures. However, global norms and international standards for biosafety and biosecurity are often highly variable between different countries and should be noted before starting with the work [83].

## 9. Patient-Derived Cell Lines, Organoids, Xenograft Models, and Conditional Reprogramming

As discussed, established cell lines can undergo genetic drift or phenotypic alterations after long-term passaging. Therefore, they may no longer faithfully represent all the molecular features that were characteristic of the initial cell type they originated as. Consequently, scientists have developed techniques that allow the establishment of primary cells from either tissue or blood from healthy donors or subjects suffering from a defined disease. For humans, these patient-derived cell lines have high translational clinical relevance [84]. However, spontaneous immortalization is commonly a rare event and the establishment of respective cells was most often performed in the past by transformation with viral oncoproteins that partially deregulate the cell cycle or by overexpression of TERT that replaces short DNA segments that are lost during cell replication and are involved in control of cell senescence [6]. Similarly, the CRISPR-Cas9-mediated targeting of oncogenes that can be used to immortalize cells in vitro has been identified as an effective tool for establishing immortalized cell lines [7,8].

However, it should be critically noted that these manipulations can exert transcriptional and cell cycle effects and, further, that the inhibition of DNA damage signaling pathways by respective agents leads to the accumulation of mutations [85,86]. More recently, conditional reprogramming (CR) has emerged as a powerful tool for the establishment of long-term cultures of primary cells [86]. The technique of CR that was first established in 2012 allows the induction of normal and tumor epithelial cells from many tissues to proliferate indefinitely in vitro. In this technique, cells can be conditionally reprogrammed by co-culturing them with irradiated fibroblast feeder cells and the Rho kinase inhibitor Y-27632 [87]. This technology is now widely used to establish patient-derived cell cultures from both normal and diseased cells. In this procedure, the epithelial cells are reprogrammed to acquire an adult stem cell character by transferring the cells from standard culture medium to a CR medium that reverses their differentiation state and allows the generation of large numbers of cells for use in patient-derived models [87]. As such, CR offers exciting possibilities in precision medicine, regenerative medicine, drug testing, gene expression profiling, xenograft studies, and to define genetic, epigenetic, and metabolic alterations occurring during the transition from a normal to a tumor cell phenotype [88]. Importantly, sophisticated protocols are now available that further allow the use of CR for the rapid and efficient expansion of non-epithelial cells including those of neural, neuroendocrine, endocrine, and mesenchymal origin that conditionally can be grown for long periods [89].

In personalized medicine, organoids, which are self-organized 3D tissues typically derived from pluripotent fetal or adult stem cells, have gained enormous interest [90]. They are a kind of miniaturized and simplified version of an organ that forms in a selective 3D medium that includes a set of growth factors [91]. In particular, patient-derived organoids (PDOs) have been widely introduced in cancer research. They recapitulate basic features of primary tumors including histological complexity and genetic characteristics and are therefore ideally useful to predict the sensitivity toward antitumor drugs or aspects of tumor progression [92].

Similarly, patient-derived xenograft (PDX) models are dynamic entities in which cancer evolution can be experimentally monitored [93]. PDXs are cancer models established by implanting and growing a patient’s tumor cells in a suitable animal host. In most cases, the recipient is an immunodeficient mouse engrafted with a human immune system [94]. These models have become a useful experimental tool for the study of molecular interactions between human immunity and cancer cells. Particularly, these models have become highly attractive in basic research to understand aspects of cancer progression and metastasis. In addition, PDXs are frequently used in preclinical cancer research to identify novel predictive cancer biomarkers, test the efficacy of cancer drugs, investigate intra-tumoral heterogeneity and clonal dynamics, evaluate personalized therapy options, and to test the general translational hypothesis [94,95].

All these advanced cell culture techniques that more closely mimic the cellular microenvironment are nowadays an integral part of basic and clinical research. However, the usage of these patient-derived models (PDMs) requires extensive expertise, training, and quality control. To foster the development of respective models, several international consortia have been established with the aim to generate novel human tumor-derived culture models with associated genomic and clinical data. Representative initiatives have been launched by the National Cancer Institute (NCI) and the Human Cancer Models Initiative (HCMI) [96,97].

## 10. Cleaning and Sterilization

Biosafety depends on the cleanness of the laboratory. The general guideline is to strictly follow all possible safety rules. Ignoring or failing to follow any safety regulations can result in laboratory-associated infections and environmental contamination. Therefore, the biological risks need to be reduced by decontamination of biological agents that were used during laboratory operations.

In proper waste management, the inactivation methods should be appropriately validated whenever possible. In principal, there are four main categories for decontamination, namely (i) sterilization by heat, (ii) disinfection with liquids, (iii) disinfection by vapors and gases, and (iv) exposure to UV radiation.

Autoclaving is the most effective and reliable method to sterilize laboratory materials and decontaminate or inactivate biological agents. It is a sterilization method that uses high-pressure water steam and is the method of choice for decontamination of culture media, glassware, and pipette tips. However, it is essential that sufficient high temperature and pressure are maintained for a period of time that also guarantees spore inactivation. Typically, autoclaving for 60–90 min at 121 °C is sufficient to achieve a waste temperature of at least 115 °C for 20 min. Moreover, the operation and maintenance of autoclaves should be performed only by trained individuals and the success of autoclaving should be regularly checked by biological indicators. It is essential that waste or materials subjected to autoclaving are placed in containers or sealed autoclave bags that permit good heat penetration [98]. Hazardous chemical substances or radioactive waste should not be autoclaved. Contaminated scalpel blades, hypodermic needles, knives, and broken glass should be collected in puncture-proof containers with covers. Large volumes of liquid waste and contaminated media should be decontaminated before disposal in the sanitary water.

Chemical disinfection is usually the preferred method for decontamination of surfaces and furniture. For the optimal effectiveness of a disinfectant, several factors have to be considered. First, a disinfectant should have specificity for the biological agent to be removed. Second, a disinfectant should be suitable for the field of application because there can be significant activity differences when applied to surfaces or liquids. Finally, disinfectants may differ in their general application conditions (e.g., required contact time, working concentration) and in their effectiveness in the presence of other influencing factors (e.g., acids, organic load).

Similar vapors and gases applied in closed systems can provide excellent disinfection. Aerosols of hydrogen peroxide, chlorine dioxide, glutaraldehyde, paraformaldehyde, ethylene oxide, and peracetic acid are used in some laboratories to decontaminate biosafety cabinets or incubators [98,99]. However, all these chemicals are hazardous and disinfection with these substances should only be conducted by experienced and trained personnel.

Finally, exposure to ultraviolet (UV) radiation can effectively destroy most microorganisms. In particular, the UV-C light (spectral range: 100–280 nm), which is absorbed by the atmosphere, has the most destructive power for a wide spectrum of microorganisms [99]. Therefore, this UV region is often used in biological safety cabinets to reduce surface contamination [98]. However, the exposure to UV can cause burns to the eyes and skin of operators and should therefore be applied only with precaution. For cell culture beginners, it is therefore advisable to receive theoretical and hands-on training by experts before using UV-C radiation devices for cleaning and disinfection.

## 11. Conclusions

Cell culture plays an important role in biomedical research. However, cell misidentification, intra- and inter-species cross-contamination, and infection by bacteria, fungi, yeast, or viruses can lead to fatal consequences and pollution of the scientific literature. Therefore, regular testing for contamination and cell authentication testing should be mandatory in each laboratory working with cell cultures. Biological contamination with bacteria, molds, and yeast can be effectively removed by diverse bactericidal or fungicidal acting components. The ICLAC register of misidentified cell lines and the linked Cellosaurus knowledge resource are extremely important knowledge resources and provide helpful search tools such as CLASTR for comparing STR profiles. In addition, the implementation of good cell culture practice and aseptic techniques are essential to increase cell culture safety, promote the generation of reproducible scientific data, and facilitate comparability of results established in different laboratories. In addition, proper staff training and standardization of documentation and reporting of cell culture procedures are further effective means to promote high-quality work and safety in a cell culture laboratory. Recent advances in the generation of more physiologically relevant PDMs such as PDOs and PDXs have revolutionized common cell culture methods and helped to better understand human biology and pathophysiology. In this regard, CR is an attractive technique that can be used to rapidly and efficiently establish patient-derived cell cultures for basic and clinical studies and further significantly substitute animal-based research. Nevertheless, it should be kept in mind that the best designed and most well engineered cell culture laboratory is only as good as its least competent worker.

## Figures and Tables

**Figure 1 cells-12-00682-f001:**
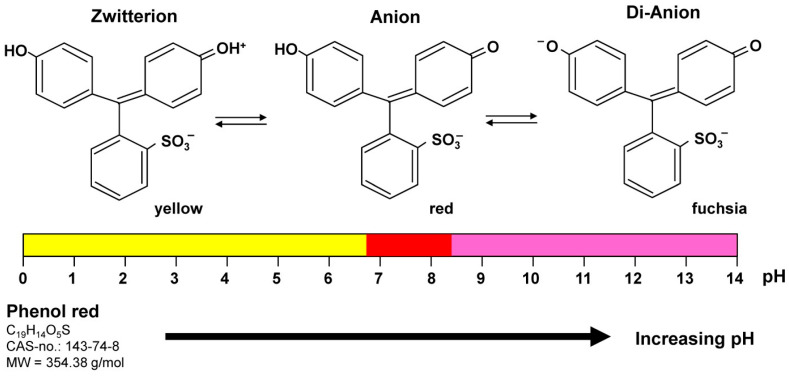
Monitoring pH changes in cell culture by phenol red. Phenol red (Phenolsulfonphthalein) is a triphenylmethane dye used in cell cultures to monitor pH changes. Its color exhibits a gradual transition from yellow to red and fuchsia with increasing pH.

**Figure 2 cells-12-00682-f002:**
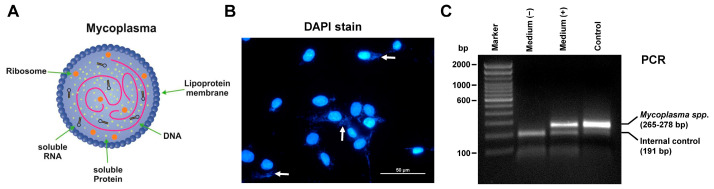
Detection of *Mycoplasma* spp. in cell culture supernatants. (**A**) Mycoplasmas are the smallest self-replicating bacteria devoid of a cell wall. They consist of a lipoprotein plasma membrane, ribosomes, and a genome consisting of a circular, double-stranded DNA molecule. (**B**) A monolayer of a mycoplasma-contaminated rat cell line was stained with DAPI and analyzed under a fluorescent microscope with an appropriate UV filter package (340/380 nm excitation) at a magnification of 400×. Please note the very small bright extranuclear dots that are typical for mycoplasmas (marked by arrows). The scale bar represents 50 µm. (**C**) In the depicted analysis, 2 µL normal control medium (Medium (−) and 2 µL medium taken from a *Mycoplasma*-infected cell line (Medium (+)) were tested for *Mycoplasma* infection using the Venor^®^GeM OneStep kit (#11-8050, Minerva biolabs GmbH, Berlin, Germany) according to the manufacturer’s instructions. As a positive control, 2 µL of the positive control included in the kit system was co-amplified. The internal control in each sample resulting in an amplicon of 191 bp in size demonstrates the functionality of the PCR reaction, while an amplicon of 265–278 bp in size indicates a contamination with *Mycoplasma spp*. The PCR products were separated on a 2% standard agarose gel containing ethidium bromide and amplicons visualized using a standard gel imager.

**Figure 3 cells-12-00682-f003:**
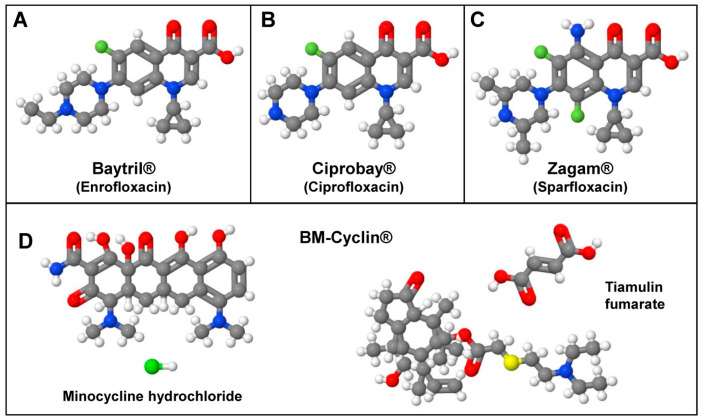
Selected compounds useful to remove or prevent *Mycoplasma* contaminations. (**A**) Baytril (CAS-no.: 93106-60-6), (**B**) Ciprobay (CAS-no.: 85721-33-1), (**C**) Zagam (CAS-no.: 110871-86-8), and (**D**) BM-Cyclin containing Minocycline hydrochloride (CAS-no.: 13614-98-7) and Tiamulin fumarate (CAS-no.: 55297-96-6) are powerful anti-mycoplasma agents that are used to eliminate or prevent mycoplasma contamination. For the biological activity of the depicted compounds, refer to Table 3.

**Figure 4 cells-12-00682-f004:**
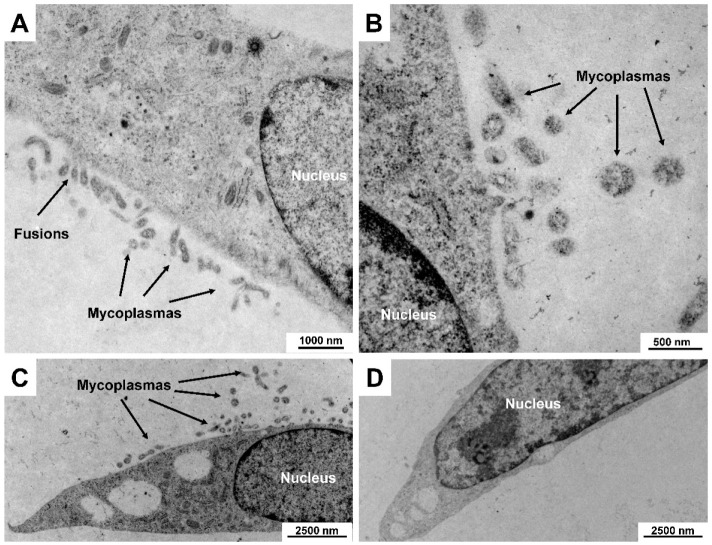
Clearance of mycoplasma contamination in a rat cell culture as assessed by electron microcopy. A contaminated cell culture was treated with Plasmocin for 2 weeks. Shown are representative electron microscopic images of cells before (**A**–**C**) and after (**D**) treatment with the broad-spectrum anti-mycoplasma reagent showing that the mycoplasmas were successfully eliminated from the culture. Images were taken at original magnifications of 4646× to 27,800×. Please note the characteristic fusions of the mycoplasma membrane with the cytoplasmic membrane of the host cell in (**A**).

**Figure 5 cells-12-00682-f005:**
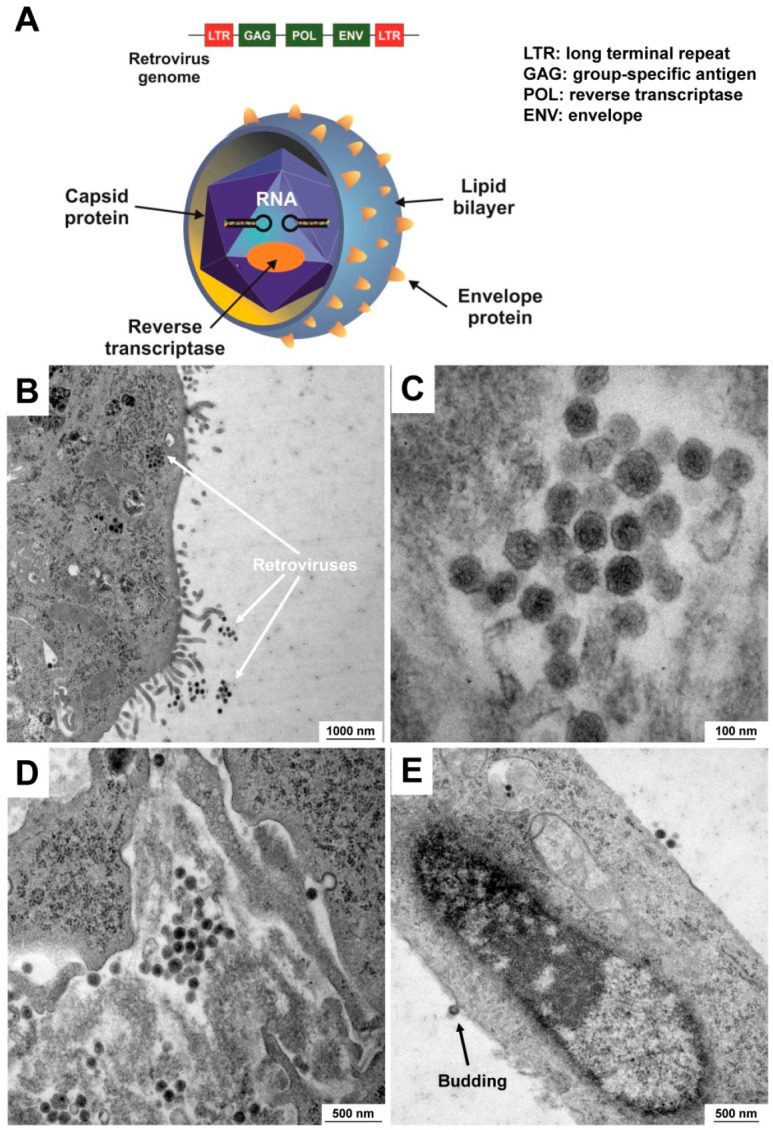
Retroviral contamination in the mouse cell line GRX. (**A**) General structure of a retrovirus. A retrovirus is an enveloped particle about 100 nm in diameter. The envelope encoded by the env gene is composed of glycoprotein that is encapsulated by lipids obtained from the plasma membrane of the host cells during the budding process. The retroviral genome further consists of two identical single-stranded RNA molecules 7–9 kb in length. Group-specific antigens (gag) are major components of the viral capsid, while the reverse transcriptase (pol) is necessary for synthesis of viral DNA. (**B**) Occurrence of a not-specified retrovirus in the mouse hepatic stellate line GRX as assessed by electron microscopic analysis. Retroviral particles can be found in the culture medium and in the cytosol. (**C**,**D**) Higher magnification images of retroviruses show the typical spherical structure. (**E**) This electron microscopic image shows a budding process in which a retroviral particle acquires its envelope by spreading through the host’s plasma membrane. Magnifications are 10,000× (**B**), 100,000× (**C**), 27,800× (**D**), and 21,560× (**E**), respectively.

**Figure 6 cells-12-00682-f006:**
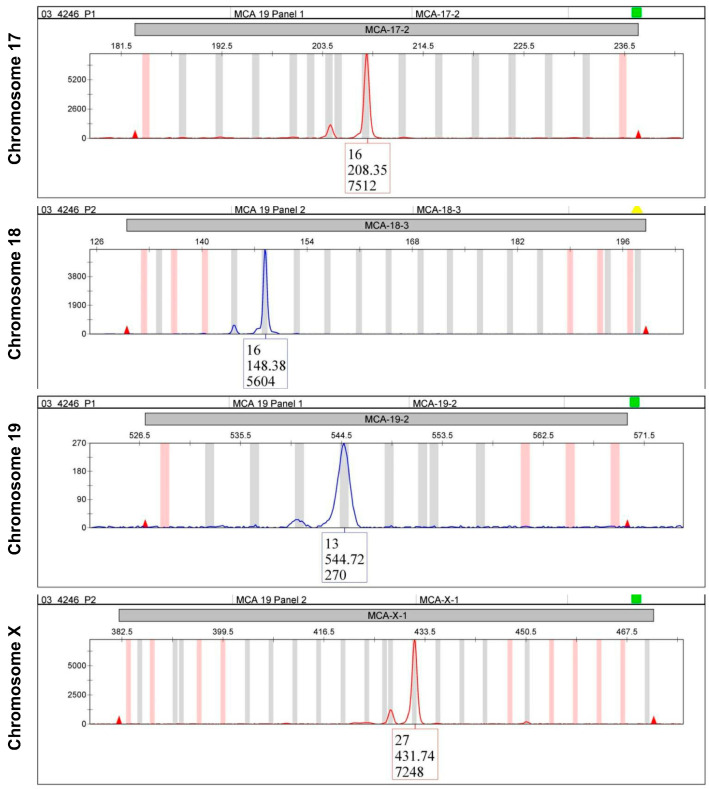
Chromatograms of typical short tandem repeat (STR) markers for a murine cell line. Genomic DNA from murine AML12 cells was isolated and subjected to STR profiling using the CellCheck^TM^ Mouse system (IDEXX, Kornwestheim, Germany). This system amplifies 19 species-specific STR markers spread on 17 chromosomes. In this figure, four representative STR profiles from chromosomes 17, 18, 19, and chromosome X are depicted. For the complete STR profile of that cell line, refer to Table 5.

**Table 1 cells-12-00682-t001:** Formulation of a Dulbecco’s modified Eagle medium (DMEM) with high glucose.

Inorganic salts/buffers: CaCl_2_: 0.2 g/L, Fe(NO_3_)_3_ × 9 H_2_O: 0.0001 g/L, MgSO_4_: 0.09767 g/L, KCl: 0.4 g/L, NaHCO_3_: 3.7 g/L, NaCl: 6.4 g/L, NaH_2_PO_4_: 0.109 g/L
Amino acids: L-Arginine × HCl: 0.084 g/L, L-Glutamine: 0.584 g/L ^1^, Glycine: 0.03 g/L, L-Histidine × HCl × H_2_O: 0.042 g/L, L-Isoleucine: 0.105 g/L, L-Leucine: 0.105 g/L, L-Lysine × HCl: 1.46 g/L, L-Phenylalanine: 0.066 g/L, L-Serine: 0.042 g/L, L-Threonine: 0.095 g/L, L-Tryptophan: 0.016 g/L, L-Tyrosine × 2 Na × 2 H_2_O: 0.12037 g/L, L-Valine: 0.094 g/L
Vitamins: Choline chloride: 0.004 g/L, Folic acid: 0.004 g/L, *myo*-inositol: 0.0072 g/L, Niacinamide: 0.004 g/L, D-Pantothenic acid (hemicalcium): 0.004 g/L, Pyridoxal hydrochloride: 0.004 g/L, Riboflavin: 0.0004 g/L, Thiamine × HCl: 0.004 g/L
Others: D-Glucose: 4.5 g/L ^2^, Phenol red × Na: 0.0159 g/L ^3^, Pyruvic acid × Na: 0.11 g/L

^1^ In some formulations, L-Glutamine should be freshly added, ^2^ the concentration of D-Glucose in DMEM with low glucose is 1 g/L, ^3^ Phenol red has a weak estrogenic activity [20] and is therefore omitted in some DMEM formulations.

**Table 2 cells-12-00682-t002:** Widespread cell culture contaminants.

Contaminant	Remarks
Viruses (*Viridae*)	Viral contamination (e.g., HIV, HBV, EBV, SHBV) is hard to detect because they do not affect cellular growth. Based on their extremely small size (~100 nm in diameter), they are not visible under a bright-field microscope. However, infections with cytopathic viruses can destroy the culture. In addition, virally infected cell cultures represent a potential health hazard for laboratory personnel.
Mycoplasmas (*Mollicutes*)	Mycoplasmas are spherical to filamentous cells with no cell walls and intracytoplasmic membranes. They are the smallest self-replicating organisms with a diameter of ~ 300 nm and small genomes (~ 500 to 1000 genes). Infection can alter the host culture’s cell functions including growth, metabolism, migration, morphology, and responsiveness towards growth factors. In addition, some mycoplasma species can provoke chromosomal aberrations and damage.
Bacteria (*Bacteriaceae*)	The shape and size of bacteria can vary considerably ranging from 0.5 to 1.0 µm up to 10 to 20 µm in spiral forms. The bacterial genomes can range from about 130 kbp to over 14 Mbp and typically consist of 500–1200 genes (parasitic bacteria), 1500–1700 (free-living bacteria), and 1500–2700 genes (archaea). Most bacterial contaminants are able to quickly colonize and the flourish in cell culture media. Respective contamination can usually be readily detected by microscopy as tiny, moving granules between the cells within a few days of initial contamination.
Yeast and mold (*Fungi*)	Yeast cells are fungi that multiply faster than mammalian cells. The typical size of yeast and mold is 3–4 µm (but can be up to 40 µm). Contamination becomes clearly obvious by microscopic analysis or color change of the medium within 2–3 days. Antibiotics such as penicillin and streptomycin have no toxic effects on yeast.
Parasites	Different intracellular protozoan parasites (e.g., *Toxoplasma gondii*, *Trypanosoma cruzi*, *Leishmania spp*., *Cryptosporidium parvum*, *Plasmodium spp*.) may be included in freshly prepared primary cell cultures originating from a donor organism that is known or suspected to be infected with respective parasites. Special safety precautions should be considered and protective clothing and equipment might be necessary. Needles and other sharp objects should be omitted when working with parasite-infected cell lines [29].
Prions	Prions are devoid of nucleic acids and consist primarily of protein termed PrPSc. Although most cell lines are resistant to prion infection, some cells lines are susceptible to prions and can promote stable and persistent replication of prions [30]. They can be included in cell culture media enriched with serum of bovine origin [31]. Prions are difficult to inactivate.
Chemical, biological, and other nonliving contaminants	Endotoxin/lipopolysaccharides, detergents, radicals, hormones, growth factors, metals, residues of disinfectants and cleaning agents, plasticizers, and other impurities can impact proper cell growth. Chemical contamination can result from contaminated reagents, water, sera or some culture additives. In addition, detergents or other deposits on storage vessels, glassware, pipettes or instruments introduced by disinfection can be sources of contamination. Plastic tubing and storage bottles can release plasticizers. Free radicals can be generated by photoactivation of tryptophan, riboflavin, or buffering agents (e.g., HEPES and PIPES) when exposed to extensive visible or fluorescent light [32].
Inter- and intra-species cross-contamination	The incidence and extent of cell line cross-contamination is rather high [33,34]. The sources of inter- and intra-species cross-contamination are manifold (e.g., spreading via aerosols, usage of unplugged pipettes, sharing media and reagents among different cell lines, usage of conditioned medium, etc.) [35].

**Table 3 cells-12-00682-t003:** Compounds used for removal of *Mycoplasma* from contaminated cell cultures.

Compound	Composition
BM-Cyclin	Tiamulin fumarate (a Macrolide) and Minocycline hydrochloride (a Tetracycline)
Ciprobay	Ciprofloxacin (a Quinolone)
Mycoplasma Removal Agent (MRA)	4-oxo-quinoline-3-carboxylic acid derivative (a Quinolone)
Plasmocin	Contains two bactericidal components (a macrolide acting on the protein synthesis machinery by inhibiting the 50S ribosomal subunit and a fluoroquinolone inhibiting the DNA gyrase)
Baytril	Enrofloxacin (a Quinolone, inhibitor of DNA gyrase)
Zagam	Sparfloxacin (a Quinolone, inhibitor of DNA gyrase)
MycoZap	Ready-to-use combination of a not-disclosed surface-active antimicrobial peptide (MycoZap reagent 1) and a not-disclosed antibiotic (MycoZap reagent 2).
MycoRAZOR	Ready-to-use antibiotic mixture prepared in PBS acting against a large variety of mycoplasma by acting on the protein synthesis mechanism by interfering with the ribosome translation of the mycoplasms as well as with their transcription apparatus.
Normocin	Three antibiotics. Two of these compounds act on mycoplasmas, Gram-positive, and Gram-negative bacteria by blocking DNA and protein synthesis. The third compound eradicates fungi, including yeasts, by disrupting ionic exchange through the cell membrane.
Fungin	The soluble form of Pimarcin, a polyene that attacks yeasts, molds, and fungi by disrupting ionic exchange through the cell membrane.
Plasmocure	It contains two bactericidal components belonging to different antibiotic families. The first antibiotic binds to the 50S subunit of the ribosome and blocks peptidyltransferase activity, while the second antibiotic binds to isoleucyl-tRNA synthetase, thereby halting the incorporation of isoleucine into bacterial proteins.
Normocure	Contains three bactericidal components belonging to different antibiotic families that inhibit DNA and protein synthesis and disrupt membrane integrity by targeting structures that are absent in eukaryotic cells.

**Table 4 cells-12-00682-t004:** Misidentified cell lines (divided according to claimed species) ^1^.

**Human (*Homo sapiens*):** ‘1.1B4; 1E8; 2008/C13*5.25; 222; 2474/90; 2563 (MAC-21); 28SC-ES; 2957/90; 3051/80; 3AB-OS; 41M; 5-8F; 6-10B; A172TR3 (U251-TR3); ACC2; ACC3; ACCM; ACCNS; ACCS; ADLC-5M2; AG-F; AKI; ALVA-31; ALVA-41; ALVA-55; ALVA-101; AO; ARO81-1 (ARO); AV3; AZ521; BCC1/KMC; BE-13; BEL-7402; BEL-7404; BGC-823; BHP 10-3; BHP 14-9; BHP 15-3; BHP 17-10; BHP 18-21; BHP 2-7; BHP 5-16; BHP 7-13; BIC-1; BLIN-1; BM-1604; BrCA 5; BSCC-93; C16; C-433; CAC2; CaES-17; CaMa (clone 15); CaOV; Caov-2; CaVe; CCL3; CGTH-W-1; CH1; CH1-cisR; Chang liver; CHB; CHP-234; Clom 15; Clone 1-5c-4; Clone-16; CMP; CMPII C2; CNDT2; CNE-1; CNE-2; CO (COLE); COLO-38; COLO-587; COLO-677; COLO-775; COLO-818; CoLo-TC; D18T; D-54 MG; D98/AH; D98/AH2 Clone B; DAMI; DAPT; DD; Det30A; Detroit-6 (Det6); Detroit-98; Detroit 98/AG; Detroit 98/AH-2; Detroit 98/AH-R; Detroit 98s; DM12; DM14; DRO90-1 (DRO); DuPro-1; E006AA; E006AA-hat; EB33; ECC-1; ECV-304; ED27; EH; EJ-1; Ej138; EL 1; ElCo; EPLC3-2M1; EPLC-65; ESP1; ETK-1; EU-1; EU-7; EUE; EVLC2; F2-4E5; F2-5B6; F255A4; FB2; FL; Flow 13000; Flow 5000; Flow 6000; Flow 7000; FQ; G-11; GHE; Girardi heart; GLC-82; GM1312; GOS-3; GREF-X; GR-M; GT3TKB; H-494; H7D7A; H7D7B; H7D7BD5; H7D7C; H7D7D; HAC15; HAC-84; HAG; HBC; HBL-100; HBT-3; HBT-39b; HBT-E (HBT-3 clone); HCC60; HCE; HCu-10; HCu-18; HCu-22; HCu-27; HCu-33; HCu-37; HCu-39; HCV-29Tmv; HEC-155; HEC-180; HEK; HEK/HRV; HEL-R66; HEp-2 (H.Ep.-2); Hep-2C; Hep2 (Clone 2B); HES; HIMEG-1; HKB-1; HKMUS; HKMUS-SF; HL111783; HMV-1; HNOS; HO-8910; HO-8910PM; HONE-1; HPB-MLT; HPC-36M; hPTC; HROBML03; Hs 677.St; HSC-41; HSG; HSG-AZA1; HSG-AZA3; HSGc-C5; HS-SULTAN; HSY; hTERT-EEC; Hu1734; Hu456; Hu549; Hu609; Hu609Tmv; Hu961a, Hu961t; HuK^o^39; HuL-1; Hut; IMC-2; IMC-3; IMC-4; Intestine 407 (Int-407, HEI); IPDDC-A2; IPRB; IPTP/98; IST-1; J-111; J96; JCA-1; JHC; JHT; JHU012; JHU013; JHU019; JHU028; JMAR; JOSK-I; JOSK-K; JOSK-M; JOSK-S; JROECL 47 (OE47); JROECL 50 (OE50); JTC-17; JTC-3; K051; K1; K2; K5; KAK1; KAT10; KAT4; KAT5; KAT50; KAT7; KB; KB-3-1; KB-V1; KCI-MOH1; KKU-213B (KKU-M214); KKU-213C (KKU-M156); KM20; KM20L2; KM-3; KM3; KMS-21-BM; KMT-2; KOSC-3; KP-1N; KPB-M15; KPL-1; KP-P1; KSY-1; KU7; KU-YS; L-02; L-132; L-41; LC5; LC5-HIV; LED-Ti; LLC-15MB; LN-319; LN-443; LR10.6; LTEP-a2; LU; LU 106; Lu-130; M10T; M4A4; M4A4 GFP; M4A4 LM3-2 GFP; M4A4 LM3-4 CL16 GFP; MA-1; MA-160; MaTu; MC-4000; McCoy; MCF-7/AdrR (NCI/ADR-RES); MDA-MB-435; MDA-MB-435S; MDA-N; MDS; MEL-HO; MEL-WIE; MGC-803; MGH-U1 (EJ); MGH-U2 (HM); MHH-225; Minnesota EE; MKB-1; MKN28; MOBS-1; MOLT-15; MPanc-96; MRO87-1; MT-1; MT-3; MUM2C; MUTZ-1; MV522; NC-37; NCC16; NCI-H1264; NCI-H1304; NCI-H1514; NCI-H157; NCI-H1622; NCI-H1870; NCI-H249; NCI-H513; NCI-H592; NCI-H60; NCI-H630; NCI-H738; NCOL-1; NCTC 2544; NCTC 3075; ND-1; NM2C5; NM2C5 GFP; NOI-90; NOK-SI; NOSE06; NOSE07; NPA’87; NS-3; OCM-1; OCM-3; OCM-8; OCUM-6; OE; OF; ONCO-DG-1; OS 187; OST; OU-AML-1; OU-AML-2; OU-AML-3; OU-AML-4; OU-AML-5; OU-AML-6; OU-AML-7; OU-AML-8; OV2008 (A2008); Ovary1847; OVMIU; P1-1A3; P1-4D6; P39/TSUGANE (P39/TSU); Panc 01.28; Panc 06.03; PBEI; PC-93; PC-MDS; PCI-22A; PCI-22B; PCI-3; PEAZ-1; PH; PH61-N; PLB-985; PPC-1; PSV811; QGY-7701; QGY-7703; QSG-7701; RAMAK-1; RB; RBHF-1; RC-2A; RED-3; REH-6; REPC; RERF-LC-MA; RERF-LC-OK; RM-10; RMUG-L; RO-D81-1; RO-H85-1; RPMI-4788; RPMI-6666; RPTC-1; RS-1; RTSG; RY; SA4; SAM-1; SAML-1; SBC-2; SBC-7; SC (28SC); SCCTF; SCLC-16H; SCLC-24H; SEG-1; SF767; SGC-7901; SH-2; SH-3; SK-GT-5; SK-MG-1; SK-N-MC; SK-OV-4; SK-OV-6; SKW-3; SLK; SLR20; SLR24; SMMC-7721; SNB-19; SNU-1958; SPC-A1; SPI-801; SPI-802; SpR; SQ-5; SR-91; SU-DHL-7; SU-DHL-9; SUNE1; SUNE2; SW-527; SW-598; SW-608; SW-613; SW-732; SW-733; T-1; T1; T-33; T404; T406; T409; T-9; Tca8113; TCO-1; TDL-1; TDL-2; TDL-3; TDL-4; TE-12; TE-13; TE-2; TE-3; TE671; TE671 Subline No.2; TE-7; TEC61; TI-1; TK-1; TMH-1; TMM; TSCCa; TSU-Pr1; Tu-138; Tu-158LN; Tu-159; Tu-167; Tu-182; Tu-212; Tu-212LN; TuWi; U-118 MG; UM-UC-2; UM-UC-3-GFP; UPES/C; UPHHJA; UTMB-460; VC312R; WiDr; WISH; Wong-Kilbourne derivative (WKD); WRL 68; WSU-ALCL; WSU-CLL; YAA; YAP; YJ; YMB-1; YMB-1-E; Z-HL16C
**Mouse (*Mus musculus*):** 1-1ras1000; 1-1src; BALB/3T3 A31-1-1; BALB/3T3 A31-1-13; Bhas42; BT-B; MOC2-10 (MOC10)
**Rat (*Rattus norvegicus*):** HAPI; RGC-5
**Horse (*Equus caballus*):** eCAS; EEK
**Cow (*Bos taurus*):** ECTC; LF-BK; LFBK-alphaVbeta6
**Mosquito (*Culicidae*)** : Aedes aegypti, Suitor’s clone; Culiseta inornata
**Black witch moth (*Ascalapha odorata*):** Ao38 (BTI-Tnao38)
**Rainbow trout (*Oncorhynchus mykiss*):** Clone 1A; D-11; R1
**Carp (*Cyprinus carpio*):** EPC
**Dog (*Canis familiaris*):** Fitz-HSA; UCDK9B1; UCDK9B2; UCDK9B3; UCDK9B4; UCDK9B5
**Mulberry tiger moth (*Lemyra imparilis*):** FRI-SpIm-1229
**Guinea pig (*Cavia porcellus*):** GPS-M; GPS-PD
**Nile tilapia (*Oreochromis niloticus*):** Hepa-T1
**White marked tussock moth (*Orgyia leucostigma*):** IPRI-OL-7; IPRI-OL-11
**Cabbage moth (*Mamestra brassicae*):** IZD-MB-0503
**Black-spotted frog (*Pelophylax nigromaculatus***): LAH1; LAH2
**Grass frog *(Rana temporaria):*** LT-1
**Rhesus Monkey (*Macaca mulatta*) & Monkey (unspecified):** MA-104; MS (Monkey Stable)
**Rabbit (*Oryctolagus cuniculus*):** MA-111
**Crayfish (*Orconectes limosus*):** OLGA-PH-J/92 (OL-J/92)
**American dog tick (*Dermacentor variabilis*):** RML-15 [RML-RSE]
**Pig (*Sus scrofa*):** SJPL
**Domestic silkmoth (*Bombyx mori*):** SPC-BM-36

^1^ For details on listed cell lines, see ILCA Register of Misidentified Cell Lines [5].

**Table 5 cells-12-00682-t005:** Analysis of 19 STR markers in three different mouse cell lines.

STR Marker	GRX	AML12	N-HCC25 *
STR 1-1 (MCA-1-1)	10	11	16
STR 1-2 (MCA-1-2)	16	13	19
STR 2-1 (MCA-2-1)	9	9	16
STR 3-2 (MCA-3-2)	14	12	14
STR 4-2 (MCA-4-2)	19.3	20.3	20.3
STR 5-5 (MCA-5-5)	15	14, 15	17
STR 6-4 (MCA-6-4)	19	15.3	18
STR 6-7 (MCA-6-7)	12	12	17
STR 7-1 (MCA-7-1)	26	29	26.2
STR 8-1 (MCA-8-1)	16	14, 15	16
STR 9-2 (MCA-9-2)	ND	15	18
STR 11-2 (MCA-11-2)	16	18	16
STR 12-1 (MCA-11-2)	16	19	17
STR 13-1 (MCA-13-1)	17	15	17
STR 15-3 (MCA-15-3)	25.3	21.3	22.3
STR 17-2 (MCA-17-2)	16	13, 15	16
STR 18-3 (MCA-18-3)	16	21	16
STR 19-2 (MCA-19-2)	12	13	13
STR X-1 (MCA-X-1)	26, 27	26	27

* Representative STR electropherograms of selected markers (i.e., STR 17-2, STR 18-3, STR 19-2, and STR X-1) of this cell line are depicted in Figure 6. ND, not determined. More genetic details on GRX and N-HCC25 cells are given elsewhere [53,68].

## Data Availability

More experimental details about original data depicted in Figure 2, Figure 4 and Figure 6 will be made available by the corresponding author upon reasonable request.

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
