# Peer review of "A Beginner’s Guide to Cell Culture: Practical Advice for Preventing Needless Problems"

_cells, 2023, doi:10.3390/cells12050682_

Round 1
Reviewer 1 Report
1. This manuscript describes the basic knowledge required for animal cell culture. It is like a technical report for beginners in cell culture. Removing mycoplasma from cell lines and misidentifying cell lines is worth reading, but other parts of this manuscript are too basic and boring.
2. If the authors wish to retain the original title and content, it is recommended to add part of the serum-free media to this manuscript.
3. Line 56. "Slow growing cell rate" should be corrected to "slow growth rate".
4. Line 59. "After monolayer formation, halted at the G0 stage." It should be corrected to "stopped at phase G0, G1 or G2 after monolayer formation". (Mao, Z., Ke, Z., Gorbunova, V., & Seluanov, A. (2012). Replicatively senescent cells are arrest in G1 and G2 phases. Aging (Albany NY), 4(6), 431.) .
5. Line 95. "liquid form" should be "liquid form".
6. Table 4. First line '1.1B4; '1.1B4; Remove duplicates and delete '.
7. Line 429. The equation for PDL should be "PDL = 3.322(log Xe – log Xb) + S".
Author Response
Dear Reviewer 1,
please find our respond to your comments in the attached pdf-File.
Regards
Ralf Weiskirchen

Reviewer 2 Report
In this manuscripts, the authors attempt to provide a brief guide for research groups beginning to work with cell culture. It is a conceptually interesting work, however, in my opinion It is not sufficiently clarified in it the novelty compared to published guides of good laboratory practices for cell culture. This is the main aspect that the authors should improve in the case in which a revised version of the manuscript is submitted.
Comments:
1. There is no information regarding the use of 3D cultures nor for cells that must be grown in suspension (non-adherent cells).
2. There is also no mention of the use of flow cytometry for the characterization of cultured cells.
3. Sometimes, trypsin is not suitable for the expansion of cultures and there are alternatives to it that should be included in the manuscript.
4. In my opinion, a detailed description regarding the equipment and minimum characteristics of cell culture laboratories should be included in the manuscript. Likewise, the cleaning and sterilization protocols are not detailed in the manuscript. These aspects are especially relevant for those researchers who are new to cell culture.
5. The authors do not discuss the cytotoxic aspects of products used for the treatment or prevention of mycoplasma infections. For example, minocycline hydrochloride is an activator of Bcl2, tiamulin fumarate blocks CD73, etc. These effects must be taken into account since they can significantly alter the experimental results.
6. DAPI or Hoesch DNA staining are highly non-specific detection methods for the detection of mycoplasma. On the other hand, the small size of these bacteria must be taken into account for their study under light microscopy.
7. On page 5, line 160, the authors state that the size of mycoplasmas is between 0.1 and 0.2 microns, which allows their identification under a standard bright-filed microscope. However, the limit of resolution of a standard microscope is 0.2 microns.
8. In the page 3, last paragraph the authors state that researcher should avoid the permanent use of antibiotics. However, the vast majority of scientific papers use antibiotics in their experimental models. This statement should be treated in greater depth and supported by bibliographical references.
9. The statements referring to the crossing of the species barrier of some contaminating organisms, the influence of origin in the behavior of cells and the benefits of the use of well-characterized cell lines (page 16, line 442-447) are not supported by bibliographical references and should be treated in more detail.
10. In my opinion, aspects related to the treatment of waste generated in the laboratory should be taken into consideration in the manuscript.
Author Response
Dear Reviewer 2,
please find our respond to your comments in the attached pdf-File.
Regards
Ralf Weiskirchen

Round 2
Reviewer 2 Report
First, I want to thank and congratulate the authors for the modifications made in the manuscript. I consider that the new version is substantially improved with respect to the previous one. As far as I am concerned, I fully understand the motivation for writing this article and I have no further problems with the revised text. Novelty remains the most important aspect to consider for the acceptance of this manuscript for publication. Nevertheless, this is something that should be judged by the editors of the journal. Again, I want to thank the authors for their education and willingness to respond to my criticism, always well-intentioned.
Author Response
Dear Reviewer 2,
once again many thanks for your encouraging words. We are happy that we could address most of your concerns.
Regards
Ralf Weiskirchen
(on behalf of all authors)